# Ameloblastoma in a Three-Year-Old Child with Hurler Syndrome (Mucopolysaccharidosis Type I)

**Mattia Di Bartolomeo** [1,†], **Arrigo Pellacani** [1,†], **Sara Negrello** [2], **Martina Buchignani** [3], **Riccardo Nocini** [4], **Gianluca Di Massa** [5], **Greta Gianotti** [5], **Giuseppe Pollastri** [2], **Giacomo Colletti** [3,6], **Luigi Chiarini** [3] and **Alexandre Anesi** [3,*]

1  Unit of Dentistry and Maxillo-Facial Surgery, Surgery, Dentistry, Maternity and Infant Department, University of Verona, P.le L.A. Scuro 10, 37134 Verona, Italy; mattiadiba@hotmail.it (M.D.B.); arrigo.pellacani@libero.it (A.P.)
2  Cranio-Maxillo-Facial Surgery Unit, University Hospital of Modena, 41124 Modena, Italy; negrellosara86@gmail.com (S.N.); giuseppe.pollastri@aou.mo.it (G.P.)
3  Cranio-Maxillo-Facial Surgery Unit, Department of Medical and Surgical Sciences for Children & Adults, University of Modena and Reggio Emilia, Largo del Pozzo 71, 41124 Modena, Italy; martina.buchignani@gmail.com (M.B.); giacomo.colletti@unimore.it (G.C.); luigi.chiarini@unimore.it (L.C.)
4  Section of Ear Nose and Throat (ENT), Department of Surgical Sciences, Dentistry, Gynecology and Pediatrics, University of Verona, 37124 Verona, Italy; riccardo.nocini@gmail.com
5  Institute of Pathology, Department of Medical and Surgical Sciences for Children and Adults, University of Modena and Reggio Emilia, Largo del Pozzo 71, 41124 Modena, Italy; gianluca.dimassa@unipr.it (G.D.M.); greta.gianotti@unipr.it (G.G.)
6  The Vascular Birthmark Foundation, Latham, NY 12110, USA
*  Correspondence: alexandre.anesi@unimore.it; Tel.: +39-059-4224552
†  These authors contributed equally to this work.

**Abstract:** Mucopolysaccharidoses (MPS) are a family of genetic diseases associated with a deficiency of alpha-L iduronidase, which causes a lack of catabolism of glycosaminoglycans (GAGs). Therefore, the accumulation of GAGs determines a wide spectrum of symptoms, typically found in a few syndromes like Hurler syndrome (HS). Among other specific manifestations, craniofacial abnormalities are crucial for the characterization of this syndrome. Ameloblastoma is a rare, benign, slow-growing, odontogenic tumor usually located in the mandible. Clear risk factors for the development of ameloblastoma remain unknown, but black patients have a fivefold increased risk. Clinically, it is characterized by a painless, variable-sized jaw swelling. Although classified as a benign tumor, ameloblastoma often has a severe clinical outcome. The most common type of ameloblastoma is the solid/multicystic/conventional one. A computed tomography scan (CT) with and without contrast is the gold standard for evaluating this kind of neoplasia. Conservative or radical surgery is the mainstay of treatment. In this case report, we described an unusual clinical assessment of conventional ameloblastoma interesting the posterior left mandible of a 35-month-old child affected by HS. This case represented a suggestive challenge both from a diagnostic and a therapeutic point of view. The patient was disease-free at 2 years' follow-up.

**Keywords:** ameloblastoma; mucopolysaccharidoses; Hurler syndrome; odontogenic tumor; odontogenic lesion; immunohistochemistry

## 1. Introduction

Mucopolysaccharidoses (MPS) are a group of genetic diseases associated with a lack of catabolism of glycosaminoglycans (GAGs). This deficit involves the accumulation of GAGs in the lysosomes and in the extracellular matrix (ECM). Different types are classified based on the enzyme involved [1]. The deficit of the alpha-L-iduronidase enzyme (IDUA gene, located on 4p16.3) is specific for mucopolysaccharidoses type I (MPS-I) [2]. The result is an accumulation of two GAGs, heparan sulfate and dermatan sulfate, which are responsible

for the clinical manifestations of the syndrome. Depending on the clinical phenotype, MPS-I is divided into:

- Hurler syndrome (HS): severe phenotype;
- Hurler–Scheie syndrome (H–SS): intermediate phenotype;
- Scheie syndrome (SS): mild phenotype [2].

The incidence of HS is estimated at approximately 1–3 in every 100 births [3–5]. Mortality used to be considerably high in the first decade of life. In fact, the accumulation of GAGs in the cardiac and respiratory tissues strongly impacts their functionality [2]. At the same time, the cerebral deposition of GAGs also leads to an important neurocognitive deficit. Other clinical manifestations also involve the musculoskeletal system, abdominal organs and sensory organs [2]. A recent therapeutic strategy consists of a bone marrow transplantation, which considerably extends life expectancy [6]. The transplanted bone marrow produces new cells, which supply the congenital deficiency of alpha-l-iduronidase. This procedure makes it possible to severely limit the development of the cardio-respiratory consequences of HS and, if performed early, is also able to reduce the neurocognitive damage [2,6].

HS is interesting from a maxillofacial standpoint for multiple reasons. First of all, children with HS have coarse facies, with a widened nose and prominent upper orbital frames, as well as possible premature fusion of the sagittal suture [2]. At the intraoral level, the accumulation of GAGs in the tongue causes macroglossia, with consequent breathing deficits and a possible anterior flattening of the hard palate. In addition, gingival hypertrophy is also often observed, which can also result in an inflammatory and fibrous reaction [7]. Regarding bone tissues, a short and broad mandible is frequently reported in the literature, together with a condylar deformation, in the context of a multiple dysostosis [2]. The activation of matrix metalloproteinases (MMPs), pro-inflammatory cytokines and chemokines seems to be at least partially responsible for the skeletal manifestations [8]. From a dental–occlusal point of view, patients with HS typically show an anterior open bite. Affected patients frequently develop atypical dental eruptions as well as morphological abnormalities of the dental elements (such as microdontia or root changes). Typically, the shape of the teeth is described as "peg-shaped" [7,9–12]. A characteristic manifestation of HS is the presence of cystic-like dentigerous lesions, which are more frequently bilateral and located in the posterior mandibular region. Radiographically, they appear as radiolucent areas, falling under enlarged dental follicles in the differential diagnosis [9]. It has been hypothesized that their development is linked to the metabolic impairment caused by the accumulation of GAGs [11]. Finally, it should be noted that alterations of the enamel are reported in about 30% of patients with HS. These include enamel hypoplasia or hypomineralization and taurodontism [11,12]. Although alterations in amelogenesis are frequent, to the best of our knowledge, this is the first case of pediatric ameloblastoma ever described in patients affected by HS.

Ameloblastoma is the most frequent benign odontogenic tumor. It is believed to derive from cells of ectodermal origin (more likely the ameloblasts) involved in the creation of dental enamel [13]. Although its development begins during childhood, from a clinical point of view, it rarely manifests itself during the first decade of life [14]. There are no overt risk factors for the development of ameloblastoma, but black patients have a fivefold increased risk, and the simultaneous presence of non-erupted teeth is often reported [15,16]. However, its clinical behavior is controversial. The fourth edition of the World Health Organization (WHO) classification of head and neck tumors of 2017 addressed several controversial issues about the nature of odontogenic lesions [17–20]. Among them there is the debate on the actual benign nature of ameloblastoma, given its characteristics of local aggression and tendency to relapse and metastasize (mainly to the lungs). It was therefore decided to reclassify the ameloblastoma into conventional, unicistic and extra-osseous/peripheral. The follicular, desmoplastic, acanthomatous, plexiform, basaloid and granular cell subtypes are described as well. The malignant counterpart was instead called ameloblastic carcinoma, showing a higher proliferative index [17,18,20,21].

Both ameloblastoma and ameloblastic carcinoma are joined by locoregional aggression that appears to be mediated by MMPs and osteoclastic activation [13]. Currently, the therapeutic gold standard is surgical excision [16,22].

Nonetheless, while ameloblastoma accounts for almost one quarter of pediatric tumors of the jaws, to the best of our knowledge, no case of ameloblastoma has ever been described before in a child with HS [23]. The present case report therefore describes a rare clinical picture, which represented a suggestive challenge from both a diagnostic and a therapeutic point of view. In addition, some possible pathogenetic hypotheses and some future management possibilities of both ameloblastoma and Hurler syndrome are described.

## 2. Case Presentation

### 2.1. Clinical and Preoperative Radiological Presentation

A 32-month-old female patient affected by HS presented to the general practitioner in November 2019 with a non-painful mandibular swelling, noticed by her mother the day before. He prescribed an antibiotic treatment, but the patient did not show any clinical response. Therefore, after 3 days, the child was taken to the pediatric emergency room. Clinically, a hard and non-painful mass was noticed in the left posterior portion of the mandible. There was not any purulent discharge, nor other signs of infection. An ultrasonographic exam of the lesion was performed, showing an ovoidal lesion of $2 \times 2.4 \times 2$ cm. The echographic exam highlighted the cystic content of the mass as an intralesional hyperechogenic signal. No signs of vascular anomalies were detected. A CT exam was prescribed and the patient was discharged.

The patient has been followed up at a referral center for MPS. She shows an omozygotic c.46_57del mutation of the IDUA gene. Her parents are first degree cousins. Due to the HS, the patient successfully underwent allogeneic bone marrow transplantation from her HLA-haploidentical sister when she was 5 months old. No oral Graft versus Host Disease (GvHD) was noticed. At the 7th post-operative day, the patient showed an important diastolic heart failure that was successfully treated. The patient is also affected by a non-compaction cardiomyopathy and a moderate-to-severe aortic valve disease. No respiratory defects were pointed out.

Common Hurler syndrome features such as hypertelorism and characteristic nose morphology were noticed (Figure 1). No macroglossia nor gingival hypertrophy were reported. Regarding the musculoskeletal apparatus, bilateral acetabular dysplasia and extra-rotation of the right foot were detected without any skull alterations. The patient also presented a third-class occlusion according to Angle's classification. The main clinical features are reported in Table 1.

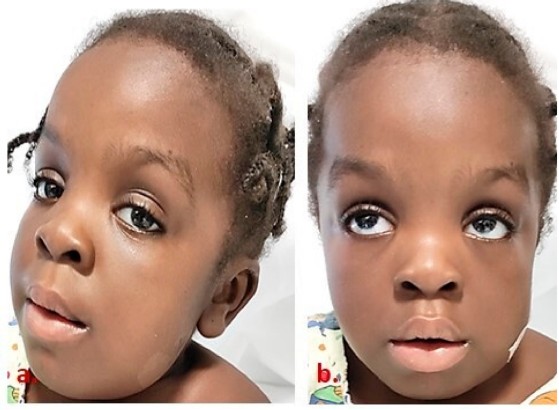

**Figure 1.** Pre-operative pictures. (**a**) Three-quarter view facing left; (**b**) frontal view. Hypertelorism and depressed nasal bridge, common features of Hurler syndrome, are clearly visible.

**Table 1.** Commonly described features in patients with MPS I, Hurler syndrome, compared to those observed in our patient.

| General Features | Present Case |
| --- | --- |
| Psychomotor impairment | Fulfilled |
| Eye disorders | Unfulfilled |
| Cardiomyopathy | Fulfilled |
| Aortic and mitral valve disease | Fulfilled |
| Hepatosplenomegaly | Unfulfilled |
| **Musculoskeletal features** | |
| Disproportionate short stature | Unfulfilled |
| Joint contractures | Unfulfilled |
| Carpal tunnel syndrome (CTS) | Unfulfilled |
| Atlanto-axial instability | Unfulfilled |
| Acetabular dysplasia | Fulfilled |
| Coxa valga | Fulfilled |
| Genu valgum | Unfulfilled |
| Trigger digits | Unfulfilled |
| **Cranio-facial characteristics** | |
| Coarse facies | Fulfilled |
| Gum hypertrophy | Unfulfilled |
| Macroglossia | Unfulfilled |
| Dysostosis multiplex | Fulfilled |
| Odontoid hypoplasia | Fulfilled |

The following month, the patient was referred a second time to the pediatric emergency room due to increased pain, fever onset and swallowing difficulties. Nonetheless no purulent discharge and no laterocervical lymphoadenomegalies were observed. No alteration of mandibular movements was noticed, either. At this time, dental elements 7.4 and 7.5 were mobile. A new ultrasonographic exam was performed in the emergency setting, showing a dimensional increase of the lesion (3.3 × 2.8 × 2.2 cm).

A CT scan was urged and then performed the following week under general anesthesia with and without contrast. The exam showed an expansive, hypodense, septate lesion of 4.0 × 3.6 × 4.0 cm in the left posterior mandibular body (Figures 2 and 3). Bilateral neck lymph nodes were enlarged due to an inflammatory reaction. The radiological frame was not diriment regarding the nature of the lesion; thus, surgical biopsies were programmed under general anesthesia.

### 2.2. Incisional Biopsy Findings

A bioptic procedure with an intraoral approach was performed under general anesthesia. The specimens showed multiple cystic structures lined by ameloblastic epithelium, together with solid foci. The diagnosis of conventional ameloblastoma was established. No perioperative complications were encountered.

### 2.3. Surgical Management

A thorough preoperative plan was made, including a cardiological evaluation and careful risk/benefit considerations. Eventually, a surgical enucleation of the tumor with an intraoral approach was proposed.

Intraoperatively, a clear surgical plane of cleavage was found (Figure 4b), which confirmed the effectiveness of the preoperative planning. The osseous defect was filled in with a fresh-frozen bone graft (Figure 4c) and the surgical approach was closed by first intention.

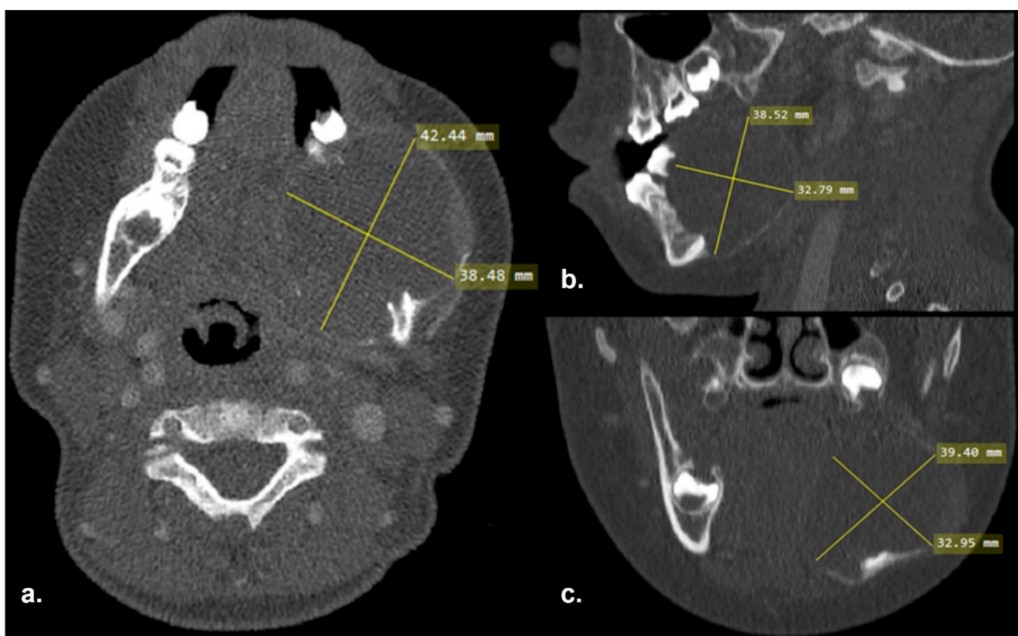

**Figure 2.** Pre-operative contrast facial computed tomography (CT) with multiplanar reconstructions showing a well-demarcated, multilocular radiolucent lesion in the left posterior mandible surrounded by a radiopaque border. The marked expansion and thinning of the buccal and lingual cortices of bone, partially eroded, and the roots resorption of the first erupted molar and the second molar germ are evident. (**a**) Axial view; (**b**) sagittal view; (**c**) coronal view.

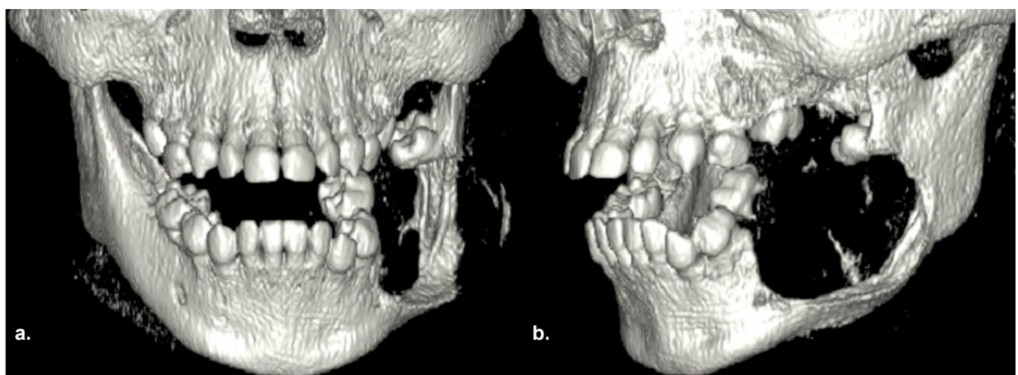

**Figure 3.** Three-dimensional facial computed tomography (CT) reconstructions. The bone defect of the left hemimandible is evident. (**a**) Frontal view; (**b**) three-quarter left-sided view.

*2.4. Conventional Ameloblastoma Diagnosis*

Gross pathological examination of the specimen highlighted a 4 × 3 × 2 cm graybrownish, capsulated lesion (Figure 5). Histologically, the neoplasm was made of both solid and cystic elements, lined by odontogenic epithelium. The islands were separated by subtle connective tissue. A thin vascularization was highlighted. Cytologically, the cells did not show significant atypia. The diagnosis of conventional ameloblastoma (with follicular pattern) was confirmed.

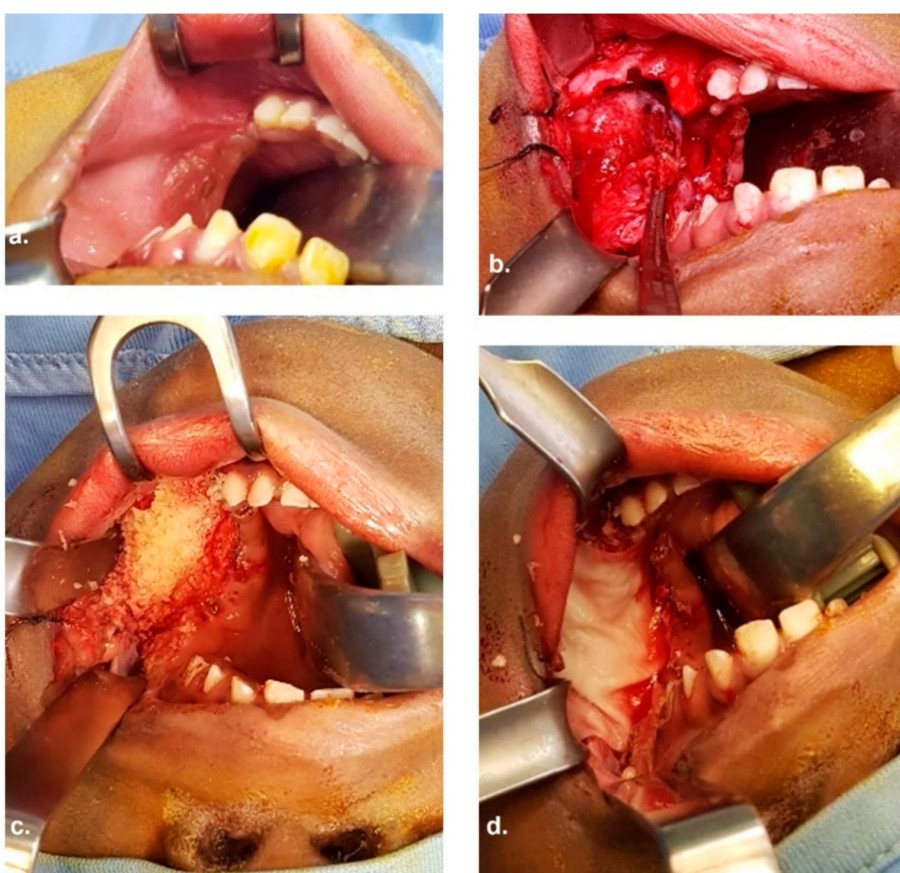

**Figure 4.** Intra-operative pictures. (**a**) Surgical field preparation; (**b**) en-bloc tumor enucleation after exposure of the body and left mandibular ramus. Note the clear cleavage plane between the fibrous tumor capsule and the bony cortical walls; (**c**) fresh-frozen bone allograft placement in the left residual mandibular bone defect; (**d**) a bilayered collagen membrane (Bio-Gide, Geistlich, Wolhusen, Switzerland) is placed to cover the bone allograft.

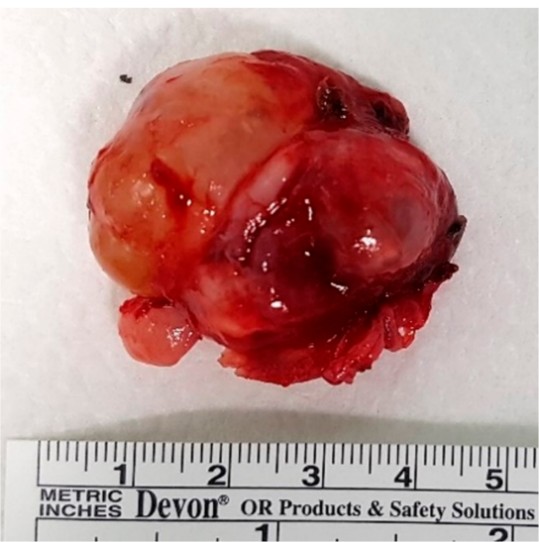

**Figure 5.** Tumor mass removed.

Immunohistochemical evaluation demonstrated strong staining for CK 5/6 and CK19, membrane staining for β-catenin and heterogenous staining for CD56 (Figure 6). Histochemically, AB-PAS positivity was highlighted inside the cystic lumina.

BRAF was focally expressed, while no expression of smooth muscle actin (SMA) or of calretinin was observed. CD34 staining showed the vascular structure of the lesion.

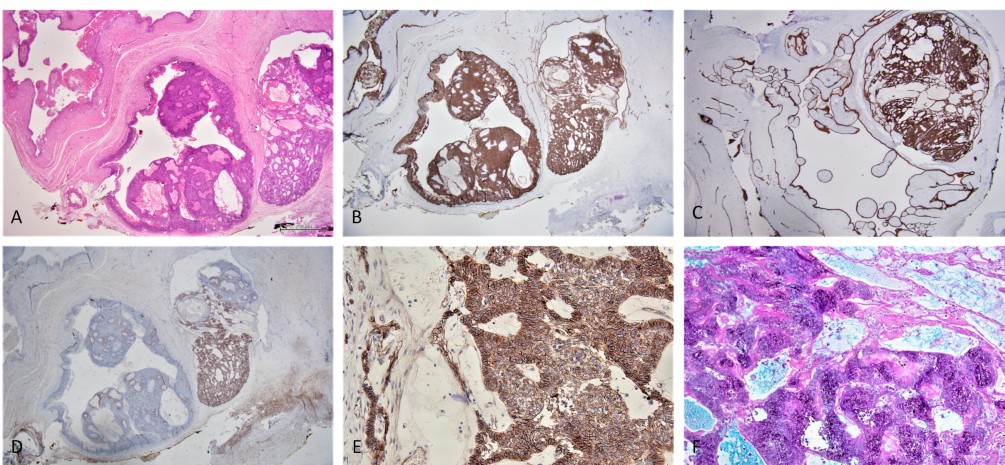

**Figure 6.** An H and E-stained section (**A**—magnification ×12.5) and immunohistochemical staining of a case of solid/multicystic/conventional ameloblastoma, showing expression of CK19 (**B**—magnification ×12.5), CK5/6 (**C**—magnification ×12.5), CD56 (**D**—magnification ×12.5), membrane expression of beta-catenin (**E**—magnification ×200) and histochemical staining AB-PAS (**F**—magnification ×100).

Some key features were noticed (Figure 7). In particular, the lesion was well capsulated in the peripheral zone, without bone cortical invasion. Moreover, diffuse expression of E-cadherin and faint staining for Ki-67 were highlighted.

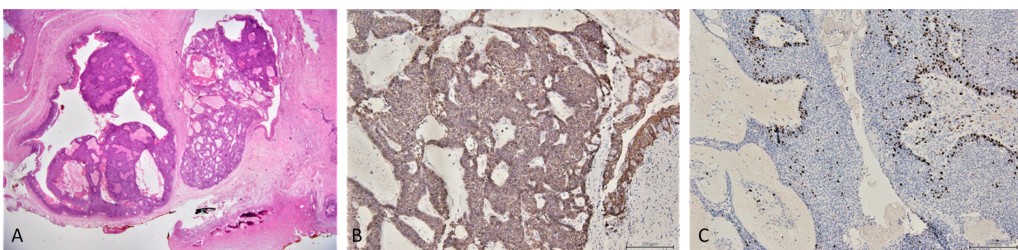

**Figure 7.** An H and E-stained section (**A**—magnification ×12.5) and immunohistochemical staining of a case of solid/multicystic/conventional ameloblastoma, showing expression of E-cadherin (**B**—magnification ×100) and Ki67 (**C**—magnification ×100).

### 2.5. Follow-Up

No postoperative complications were observed and the patient was discharged after five days. A monthly follow-up was made in the first postoperative period. A six-month postoperative CT scan was performed to make sure that the bone graft was healing well. The radiological exam did not show any recurrence of the tumor, and an exuberant bone regeneration in the left posterior mandible was observed (Figure 8). A follow-up every six months was then performed. Sixteen months after the surgery, an orthopantomography confirmed the previous findings (Figure 9). Clinically, no facial asymmetry nor paresthesia of the inferior alveolar nerve were detected.

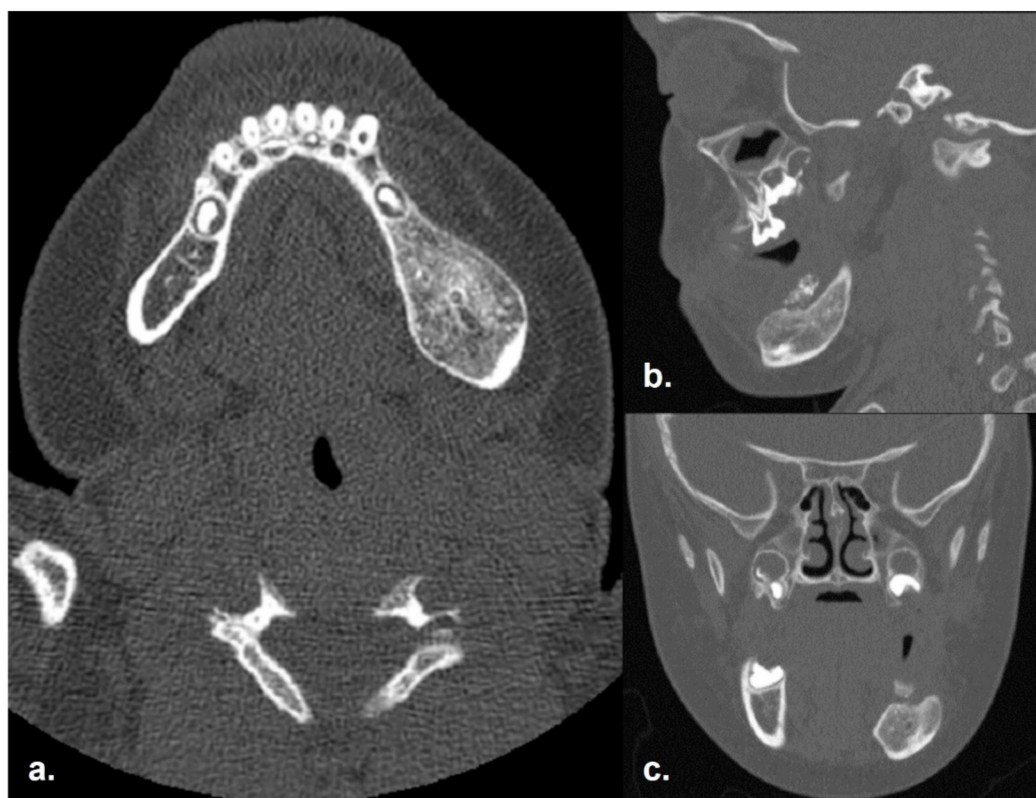

**Figure 8.** Postoperative non-contrast facial computed tomography (CT) performed 5 months after surgery. In the axial view (**a**), the asymmetry of the left mandibular body due to the exuberant bone regeneration is evident, but any bone defect is no more detectable. (**b**) Sagittal view; (**c**) coronal view.

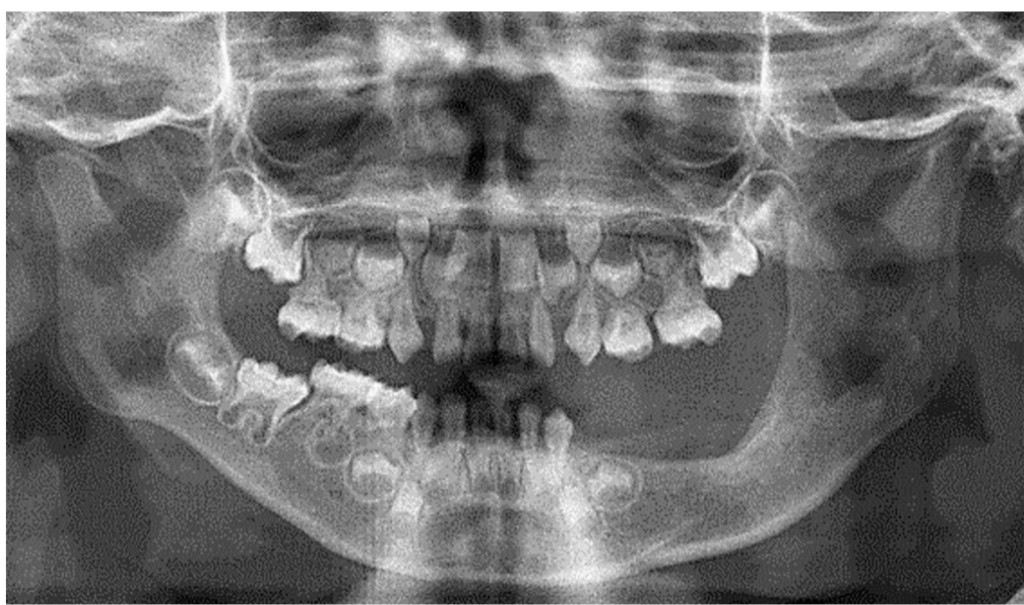

**Figure 9.** 15-month postoperative Rx orthopantomography showing excellent bone tissue regeneration at the surgical site. Bone remodeling resulted in acceptable mandibular symmetry.

The patient has been disease-free for two years as of today (Figure 10) and she is continuing her follow-up at the referral center for mucopolysaccharidoses.

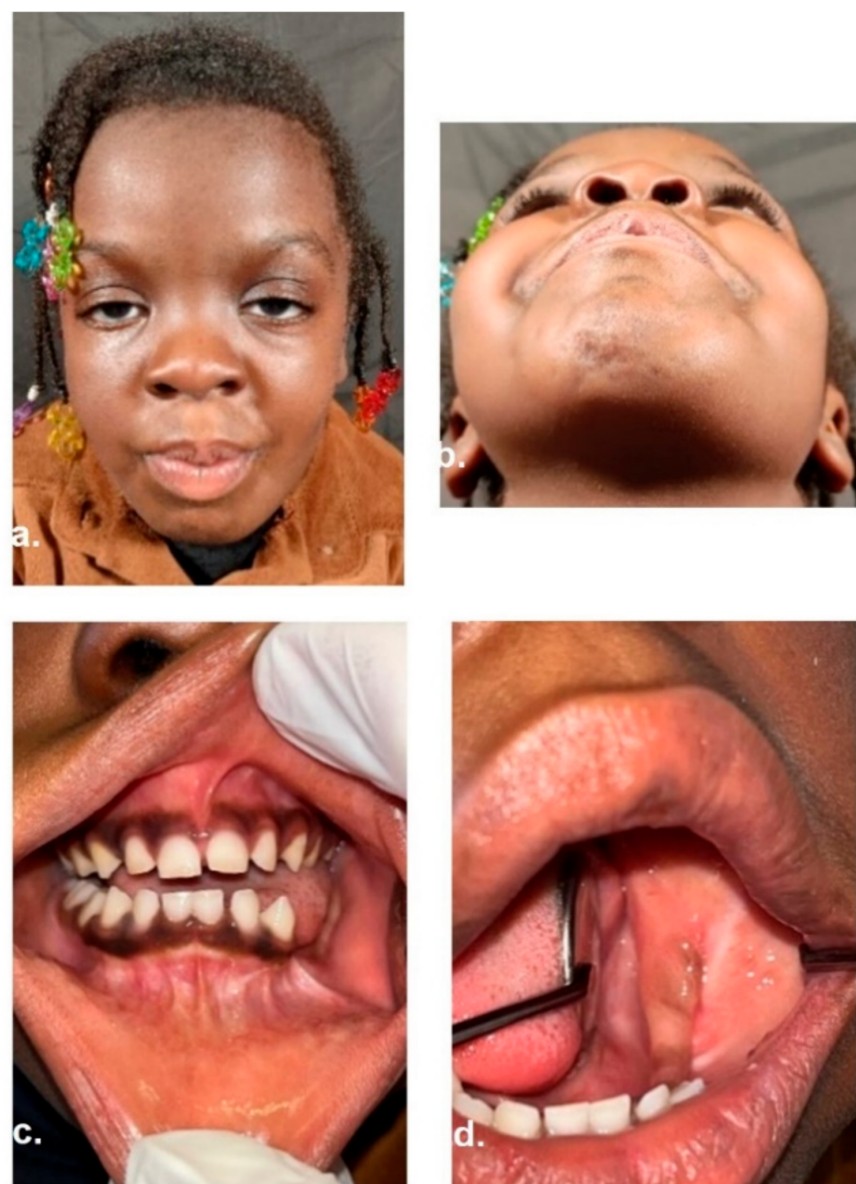

**Figure 10.** 22-month postoperative clinical condition. (**a**) Frontal view; (**b**) bottom-to-top view; (**c**) patient's occlusion; (**d**) intraoral picture.

### 3. Discussion

It is important to firstly consider the anesthesiologic management of patients with Hurler syndrome, because cardiovascular and respiratory comorbidities can be present even from a young age. Typically, the accumulation of GAGs in the heart leads to clinically relevant valvular and myocardial abnormalities [2]. Specifically, this little girl has had non-compaction cardiomyopathy and mild-to-moderate aortic insufficiency. In addition, patients with HS might have a typical phenotype with a short and not very mobile neck, ankylosis of the temporomandibular joints, macroglossia, tonsillar hypertrophy and narrow nasopharyngeal diameters [24,25]. These characteristics might result in a very challenging anesthesiologic management.

As reported by many studies, the early allogeneic transplantation might have contributed to the absence of specific characteristics considered typical of HS patients, such as gingival hypertrophy or macroglossia [11], but not present in this clinical case. Indeed, the absence of macroglossia and severe HS features certainly allowed an easier intubation and a smoother airway management in this little child. In fact, no perioperative

complications occurred either when the biopsy and resective surgeries were performed, nor when she underwent the preoperative CT. This is relevant, since the incidence of complications regarding the respiratory tract can be as high as 57% in patients with a severe phenotype, and the intubation can fail in up to 3% of cases [26]. Furthermore, the absence of macroglossia and of gingival hypertrophy in this patient led to a better intraoral surgical access to the mandibular body during surgery. The neurocognitive sphere must also be considered, since it is often impaired in HS patients [2]. Long-duration general anesthesia might affect neurocognitive development in children, as already reported by an FDA alert [27,28]. Consequently, great caution should be used when performing general anesthesia in pediatric patients.

Ameloblastomas account for nearly 25% of jaw tumors, with a global incidence of 0.5 cases per million persons per year [22]. However, it is a rare entity in childhood. The atypical age of ameloblastoma onset associated with HS certainly complicated the diagnostic pathway in this case. The new 2017 WHO classification divided ameloblastomas into conventional, unicystic, and extra-osseous/peripheral [17,18,20].

Ameloblastoma typically arises in the posterior area of the mandible and it manifests as a slow-growing lesion that does not cause any pain or paresthesia [22]. In our case, the location and the pattern of growth of the tumor occurred compatibly with the literature. On the other hand, the onset of pain is a rare manifestation in ameloblastomas. Furthermore, no meaningful correlation has ever been described between bone marrow transplantation and the development of ameloblastoma. Since the diagnosis of an ameloblastoma in a patient with HS has never been reported before, and since pain is not a typical symptom, the first diagnostic hypothesis leaned towards a cystic odontogenic lesion. Considering the above and the radioprotective precautions in children, an ultrasound examination was the diagnostic imaging investigation performed first. Afterwards, to properly assess the growth of the lesion within the mandible, it was decided to prescribe a CT under general anesthesia in agreement with the pediatricians. The CT is the diagnostic gold standard imaging investigation for mandibular bone, but in this case, it was not assertive in the definitive diagnosis. It is hoped that the use of artificial intelligence in the future might increase the radiological diagnosis of ameloblastomas [29]. It is also emphasized that it is important to share the radiological databases and the algorithms used with these technologies, both to improve the current knowledge when creating common databases and to allow the reproducibility of the experiments [30].

Despite the COVID-19 pandemic, the severe pain and the progressive growth of the mass required prompt assessment and treatment planning [31]. Therefore, the patient underwent a bioptic sampling under general anesthesia, which finally ascertained the nature of the lesion: conventional ameloblastoma (previously known as solid/multicystic). The histological finding needed some surgical considerations in treatment planning considering pediatric patients. In fact, ameloblastoma is a local, aggressive tumor, which may metastasize in up to 2% of cases (more frequently in the lungs) [22]. The therapeutic gold standard is surgical removal, which must be appropriately planned considering various factors. As a result, in a pediatric patient, a symptomatic and enlarged ameloblastoma absolutely has to be removed. This tumor can affect both the physiological eruption of teeth and bone growth [7,9,11,12,32]. This aspect is even more important in patients with HS, since it is well known that dental eruption is usually compromised and the growth of facial bones is frequently altered. Furthermore, noble structures, such as the inferior alveolar nerve, must be preserved. In this case, the use of artificial intelligence techniques to easily identify the mandibular canal may play a key role in surgical planning [33].

On the other hand, a dilemma regarding the best reconstruction strategy was raised. Several treatments were reported in the literature, including bone resection and enucleation. However, no significant differences were observed in terms of surgical outcomes [34]. An en-bloc resection with large margins of radicality is the preferred solution in adults [35]. In the pediatric setting, a substantial osseous respective deficit (through and through mandibular defect) followed by a free flap reconstruction can significantly compromise the

skeletal growth. Then, we decided to enucleate the tumor mass. As shown in Figure 4b, it was possible to intraoperatively find a clear surgical plane of cleavage. A fresh-frozen bone allograft was placed to reconstruct the bony defect. Although the bone-healing mechanisms are still under investigation, this graft is considered a safe and effective procedure [36,37]. The mandibular body appears completely ossified, as can be observed in the sixteen months' post-operative OPT. To date, there is no clinical or radiological evidence of disease recurrence.

The link between the pathogenesis of ameloblastoma and the onset of dentigerous lesions in HS can be of particular interest. First of all, some kinds of alterations in amelogenesis are reported in about one-third of the patients with HS [11,12]. Moreover, dental eruption disorders and hypodontia are very frequent (the permanent second premolar and second molar are typically involved) [11]. Although it is not a clear risk factor for ameloblastoma, an association between this odontogenic tumor and impacted teeth has been widely described. Starting from approximately three years of age, the development of endosseous cystic-like lesions such as dense fibrous connective tissue in the jaws has been reported in HS patients [9]. A typical accumulation of GAGs is also reported [11]. However, in our patient, these features did not occur and the early bone marrow transplantation could have played a crucial preventive role for these specific features. In contrast, we observed AB-PAS positive matrix within the cystic lumens (Figure 6F), suggesting its mucin nature. Regarding the pathogenesis and the growth pattern, both ameloblastomas and cystic-like lesions appear to be related to the activation of MMPs and to the activity of TNFa [13,38]. In particular, a greater peripheral proliferation was observed in ameloblastomas, with an increased osteoclastic activity mediated by RANKL and TNFa.

While the immunohistochemical diagnosis of odontogenic lesions is hardly conclusive, we found it interesting to investigate it in this specific case [39]. A typical diffuse CK5/6 and CK19 expression and membrane β-catenin positivity were documented. Interestingly, the diffuse expression of E-cadherin and the faint staining for Ki-67 were observed. These features, in association with the absence of bone cortical invasion, were related to a lower risk of recurrence of ameloblastomas [34,40,41].

Conventional ameloblastoma is the most frequent histotype and the one with the highest recurrence rate [35]. This justifies a strict follow-up. A clinical examination every six months and a yearly orthopantomography investigation were planned. The skeletal and dental abnormalities frequently seen in HS patients demand close monitoring as well [9].

The patient's parents and pediatricians were fully informed about the recurrence possibility, and the importance of an early diagnosis of tumor relapse was stressed.

It is also notable that great advances have been made regarding the knowledge of the molecular pathway involved in the development of ameloblastoma. As a matter of fact, recent works showed the association between this tumor and alterations in the MAPK/ERK pathway [42]. Mutations of the BRAF (V600E) and MEK genes have been frequently observed as well, together with those of FGFR-2 and SMO [43,44]. Targeted therapies, such as vemurafenib and dabrafenib (inhibitors of BRAF gene), trametinib (inhibitor of MEK gene), cyclopamine (inhibitor of SMO gene), ponatinib and regorafenib (inhibitors of FGFR2 gene), seem to be an encouraging upcoming option [42,43,45]. Hence, a weighted risk/benefit assessment of pediatric ameloblastoma surgical resection needs to be further defined.

## 4. Conclusions

Ameloblastoma is the most common benign odontogenic tumor, with an estimated overall incidence of 0.5 per million population per year. However, it is a rare nosological entity in children. This often implies greater diagnostic difficulties.

Moreover, the coexistence of a clinically complex syndrome may result in a real challenge in terms of identification, with additional considerations to make for the treatment planning.

To the best of our knowledge, this is the first case of pediatric ameloblastoma in a Hurler syndrome patient.

Given the potential cosmetic–functional sequelae of this locally aggressive tumor, matched with the possible systemic complications of mucopolysaccharidosis, close multidisciplinary treatment planning and follow-up are mandatory in these patients.

**Author Contributions:** Conceptualization, M.D.B., A.P. and A.A.; methodology, S.N. and M.B.; software, A.P.; validation, A.A., G.P., G.C. and L.C.; formal analysis, R.N., M.B. and G.D.M.; investigation, M.D.B., A.P. and G.G.; data curation, G.D.M., G.G. and R.N.; writing—original draft preparation, S.N., M.D.B. and M.B.; writing—review and editing, A.P., G.C. and G.P.; supervision, A.A. and L.C. All authors have read and agreed to the published version of the manuscript.

**Funding:** This research received no external funding.

**Institutional Review Board Statement:** Due to the retrospective nature of this study, it was granted an exemption by the Institutional Review Board of the University Hospital of Modena, Italy. All procedures performed involving the human participant were in accordance with the 1964 Helsinki Declaration and its later amendments or comparable ethical standards.

**Informed Consent Statement:** Informed consent was obtained from all subjects involved in the study.

**Data Availability Statement:** Not applicable.

**Acknowledgments:** We thank Luca Fabbiani for the precious technical support.

**Conflicts of Interest:** The authors declare no conflict of interest.

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
