# Peer review of "Ameloblastoma in a Three-Year-Old Child with Hurler Syndrome (Mucopolysaccharidosis Type I)"

_reports, doi:10.3390/reports5010010_

Round 1
Reviewer 1 Report
The manuscript by Di Bartolomeo et al describes an important aspect of MPS I HS. This patient was treated with HSCT at the age of 5 months.
The focus of the case is on a mandibular growth, which has never been described before in MPS conditions. The diagnostic work-up, perisurgical assessment and 2 year follow up are described. The scans and histopathological images add additional information to the case.
The manuscript is well written with a detailed discussion and thorough referencing.
Minor comment- the authors state that ameloblastoma is rare in children and in general population, and never described in MPS condition before.
Could the author comment whether there is any link with her previous HSCT and the tumour?
Author Response
Modena 3rd March 2022
Dear Editor,
Dear reviewers
Enclosed please find the revised version of our manuscript (manuscript number: reports-1598623) titled:
Ameloblastoma in a three-year-old child with Hurler Syndrome (Mucopolysaccharidosis Type I)
First of all we would like to thank the reviewers for the appropriate and useful observation, which deserve specific answers.
Reviewer 1
The manuscript by Di Bartolomeo et al describes an important aspect of MPS I HS. This patient was treated with HSCT at the age of 5 months.
The focus of the case is on a mandibular growth, which has never been described before in MPS conditions. The diagnostic work-up, perisurgical assessment and 2 year follow up are described. The scans and histopathological images add additional information to the case.
The manuscript is well written with a detailed discussion and thorough referencing.
Minor comment- the authors state that ameloblastoma is rare in children and in general population, and never described in MPS condition before.
Could the author comment whether there is any link with her previous HSCT and the tumour?
The present sentence has been added to the “Discussion” section:
Furthermore, no meaningful correlation has ever been described between bone marrow transplantation and the development of ameloblastoma.
Reviewer 2
Dear authors,
The manuscript is very interestingly. I strongly suggest this manuscript should be accepted for publication in "Reoprts" after minor revision. My comments were described as followings:
- Please check and update the references.
The references have been checked and updated. Three more references have been added.
- Before for publication, authors should consult an English language editor and write in an "academic English" as well as "must" provide proof of a certificate of editing.
The paper has been revised and edited by a qualified translator and interpreter. The certificate of editing is here attached.
Academic Editor Comments
It is a very interesting case. However, It is necessary to confirm whether the permission of the patient's parents has been obtained for the publication of this case.
The permission of the patient’s parents has been obtained and it has already been sent to the Assistant Editor, Ms. Bianca Soceanu.
We look forward to hearing back from you at your earliest convenience.
Sincerely
Alexandre Anesi, MD
Alexandre Anesi, MD
Department of Medical and Surgical Sciences for Children & Adults, Cranio‐Maxillo‐Facial Surgery, University of Modena and Reggio Emilia, Largo del Pozzo 71, 41124 Modena, Italy;
Phone +39 059 4224552
E-Mail alexandre.anesi@unimore.it
Reviewer 2 Report
Dear authors,
The manuscript is very interestingly. I strongly suggest this manuscript should be accepted for publication in "Reoprts" after minor revision. My comments were described as followings:
- Please check and update the references.
- Before for publication, authors should consult an English language editor and write in an "academic English" as well as "must" provide proof of a certificate of editing.
Author Response

(The authors gave the same response as above.)
